# Comparison of Melting Processes for WPC and the Resulting Differences in Thermal Damage, Emissions and Mechanics

**DOI:** 10.3390/ma15093393

**Published:** 2022-05-09

**Authors:** Sebastian Wiedl, Peter Karlinger, Michael Schemme, Manuela List, Holger Ruckdäschel

**Affiliations:** 1Department of Plastics Technology, Faculty of Engineering Sciences, Campus Rosenheim, Technical University of Applied Sciences Rosenheim, Hochschulstraße 1, 83024 Rosenheim, Germany; peter.karlinger@th-rosenheim.de (P.K.); michael.schemme@th-rosenheim.de (M.S.); 2Department of Environmental Technology, Faculty of Chemical Industry and Economics, Campus Burghausen, Technical University of Applied Sciences Rosenheim, Robert-Koch-Straße 28, 84489 Burghausen, Germany; manuela.list@th-rosenheim.de; 3Department of Polymer Engineering, University of Bayreuth, Universitätsstraße 30, 95447 Bayreuth, Germany; holger.ruckdaeschel@uni-bayreuth.de

**Keywords:** biocomposites, polypropylene, wood fibers, compounding methods, thermal damage, volatile organic compounds, mechanics, fiber length

## Abstract

The necessity for resource-efficient manufacturing technologies requires new developments within the field of plastic processing. Lightweight design using wood fibers as sustainable reinforcement for thermoplastics might be one solution. The processing of wood fibers requires special attention to the applied thermal load. Even at low processing temperatures, the influence of the dwell time, temperature and shear force is critical to ensure the structural integrity of fibers. Therefore, this article compares different compounding rates for polypropylene with wood fibers and highlights their effects on the olfactory, visual and mechanical properties of the injection-molded part. The study compares one-step processing, using an injection-molding compounder (IMC), with two-step processing, using a twin-scew-extruder (TSE), a heating/cooling mixer (HCM) and an internal mixer (IM) with subsequent injection molding. Although the highest fiber length was achieved by using the IMC, the best mechanical properties were achieved by the HCM and IM. The measured oxidation induction time and volatile organic compound content indicate that the lowest amount of thermal damage occurred when using the HCM and IM. The advantage of one-time melting was evened out by the dwell time. The reinforcement of thermoplastics by wood fibers depends more strongly on the structural integrity of the fibers compared to their length and homogeneity.

## 1. Introduction

For a bioeconomy based on renewable raw materials, it is essential to develop resource-saving processes, materials and applications, as well as to replace currently used environmentally unfriendly solutions. To date, lightweight constructions have been used to save resources in the manufacturing of components and their application. Fiber-reinforced plastics in particular have been used successfully in lightweight constructions [1]. However, there is a need for new developments in raw materials. The production of petroleum-based plastics and inorganic fibers made of glass has an enormous environmental impact [2,3]. Sustainable alternatives, such as flax, already exist, but have so far only been established in niche applications due to problems such as availability [4]. A good alternative is offered by wood fibers, which have similar properties, are based on domestic raw materials [5] and, compared to agricultural-based fibers, are available on a commercial scale [6]. However, a major problem of natural products such as wood, especially in the form of fibers, due to their large surface areas [7,8], is their thermal stability [9]. The sensitivity to temperature of natural products is one of the most significant obstacles to their widespread use within industry [10,11].

Various publications provide a good overview of how increased thermal loads damage the structure of wood, increase emissions (including the release of acids) and decreases its mechanics. Solid wood starts to thermally degrade at approximately 180 °C, starting with the degradation of hemicellulose [12,13]. In the case of fibers and particles, the thermal damage already starts at lower temperatures due to the larger surface area. In the studies cited, the authors present increased thermal stress for different types of wood [14,15,16,17,18,19].

Similar effects have been documented for wood when it is compounded with plastics. Due to the effects of the increased shear and thermal load, respectively, on the temperature and dwell time, wood plastic compounds (WPCs) become darker, indicating thermal degradation [20,21,22,23,24,25,26]. Different studies processed WPC up to 230 °C and reported adverse effects of increasing the processing temperature on the mechanical properties [20,21,22,23], whereas another study showed limited or no effects [24]. Increased mixing time and shear enhanced the process of degradation [25,26].

The Agency for Renewable Resources (Fachagentur Nachwachsende Rohstoffe e.V.) states that the WPC market is dominated by the use of wood flour, while wood fibers are mostly not considered [27]. The physiological differences between wood flour and wood fibers can be seen in Figure 1. Wood flour consists of spherical particles with a length and width of around 300–400 µm [27,28]. By contrast, the lengths of wood fibers and thermomechanical pulp fibers may vary from 20 µm to 4500 µm, while the fiber diameters may vary from less than 1 µm to 80 µm [29]. This results in aspect ratios from 24 to 72, as observed by Mertens et al. [29].

The findings of Thomason and Vlug, suggesting a positive correlation between mechanics and fiber length [30,31,32], have also been shown for natural fibers [33] and wood fibers [20,34,35]. Low aspect ratios cause stress concentrations that lead to decreased strength compared to neat polymers [36,37]. Not only is the length of the fiber decisive for the mechanics of the injection-molded part, but the homogeneity of the nature-based reinforcement within the polymer plays an important role as well [38,39]. Wood flour composites may display similar tensile strength properties and higher tensile moduli in comparison to wood-fiber composites [39,40]. These small-sized particles integrate well into the processing equipment and result in more homogenous materials due to their low aspect ratio and, therefore, good dispersibility within the polymeric matrix [26,40,41].

A review of the literature shows that the processing method, either injection molding or extrusion, influences the properties of the WPC part. High-density components, which are usually produced with injection molding, may result in high mechanical performance as they provide more dense contact between the matrix and wood particles [42,43]. Migneault achieved a higher density with extruded profiles but still obtained better tensile properties with injection-molded parts as the fiber orientation was more pronounced [34]. It should be considered that the fiber morphology changes throughout processing due to fiber–fiber interactions, shear forces and high temperatures [37].

Although different processing methods for wood-fiber plastic compounds have been thoroughly investigated [34,39,42,43,44,45,46], only Kahr, Specht and Stadlbauer examined different compounding methods and their effects on the resulting compounds [45,46] and components [39,46]. Kahr compared the co-rotating and counter-rotating twin-screw-extruder (TSE) with the heating–cooling mixer in terms of energy consumption [45]. Stadlbauer reviewed the co-rotating TSE with subsequent injection molding and the internal mixer with subsequent compression molding and milling of the test specimen to determine its mechanical properties. The processing method through TSE resulted in the best mechanics due to the fiber orientation, resulting density and homogeneity [39]. The fiber-length degradation was only investigated for the TSE with subsequent injection molding; the thermal degradation and resulting emissions were not investigated. None of the studies cited above integrated one-step processing by means of an injection-molding compounder (IMC) into their investigations.

This article compares four different processing rates with regard to their different degradation effects during processing. It aims to identify the least harmful processing method for WPCs that preserves the structural integrity of the wood fiber and, thus, exploits the potential of sustainable reinforcing fiber wood in the best possible way.

For this purpose, the processing of thermoplastics needs to be understood and its effects on WPCs qualitatively compared. Melt processing can be defined as a thermomechanical process in an oxygen-deficient atmosphere. The polymer is subjected to a mechanical load induced by shear. In addition, it is exposed to thermal stress, which is caused externally by heated plasticizing systems and internally by generated shear energy. The mechanical load induced by the shear further defibers and shortens the wood filler. To determine the correlation between the melting process and the mechanical load, the wood fibers extracted from the injection-molded part were analyzed via an optical measuring system, as in various publications listed in Mertens et al.’s review [29]. The thermal load led to the degradation of the wood plastic compound. The degradation was determined relatively over all the processing rates investigated. In the case of wood fibers, the oxidation induction time (OIT) served as a parameter for determining the thermal damage. If the compounding and injection molding were carried out gently, the measured OIT on the component dropped, contrary to the usual interpretation. The components of the wood that led to earlier onset of the oxidation reaction, such as hemicellulose, had not degraded at this point and, therefore, appeared in the measurement of the OIT. When increased thermal damage occurs, wood releases hydrocarbons, alcohols and organic acids [14,15] and the WPC becomes discolored and grows dark [20,21,22]. Accordingly, a measurement of the VOCs via headspace gas chromatography was performed and the colors of the injection-molded WPC parts were analyzed. Additionally, the mechanical properties of the WPC depend on both the fiber length in the component [33] and the amount of thermal energy induced [20,21,22,23]. Therefore, the basic mechanics were determined via a tensile test.

## 2. Materials and Methods

### 2.1. Materials

A polypropylene (PP) homopolymer (HJ120UB) from Borealis was chosen as the matrix material. The manufacturer specifies an MFR of 75 g/min (230 °C/2.16 kg) and a density of 0.905 g/cm^3^. Furthermore, the manufacturer assigns the material’s basic stabilization and low emissions. The spruce fibers used in the study were produced in the technical center of the Rosenheim University of Applied Sciences. A polypropylene copolymer grafted with maleic anhydride (SCONA TPPP 8112 GA) from BYK Additives was used as a coupling agent between the PP and the wood fibers, hereafter also called MAH-PP. The same mixing ratio of PP at 67 wt.-%, spruce fibers at 30 wt.-% and coupling agent at 3 wt.-% always served as the basis for the subsequent tests. The spruce fibers were produced in a thermo-mechanical process (TMP) on a 12-inch laboratory refiner type 12 1CP from Andritz AG. After defibration, the wood fibers were stored open for more than 12 months. The wood fibers used in this study were between 90 µm and 9050 µm long, giving an average fiber length of (1220 ± 210) µm. The diameter varied between 30 µm and 480 µm, giving an average fiber diameter of (48 ± 2) µm. This resulted in an aspect ratio of 24.5. High values of length and diameter refer to fiber bundles, which can be seen in Figure 1. These were a small proportion of the measured fibers (<1%).

### 2.2. Methods

#### 2.2.1. Production of the Wood-Fiber–Polypropylene Compound

The following section compares four different ways of compounding the materials mentioned in Section 2.1 Materials. All melting processes investigated are schematically illustrated in Figure 2. In all methods used, the wood fibers and wood-fiber pellets were dried at 105 °C for 3 h in a circulating air oven before further processing. After drying, the moisture content of the wood fibers was below 0.5 wt.-%.

An injection-molding compounder utilizing a twin-screw extruder was used to investigate two different processing rates (L/D = 41; D = 25 mm, type KM 300 CX IMC, KraussMaffei Group GmbH, Munich, Germany). Spruce fiber pellets were compounded with PP and coupling agent. The wood fiber pellets were fed via a side feeder after a length of 28 times their diameter (28 D) at 180 °C. For the first method used, the melt was conveyed into the shotpot injection unit and further injected into the mold. This method represents the one-step process and is hereafter called “**1-step IMC**”. This direct compounding process has already been investigated by Obermeier et al. [47].

For the second method, the melt was not processed further but extruded as a compound strand. The melt strand was cooled in a water bath, granulated in sizes suitable for metering and subsequently processed on an injection-molding machine as a second processing step. This method represents the two-step process by means of a twin-screw extruder and is referred to as “**2-step TSE**”. The initial situation (screw configuration, temperature, pressure) was identical to the one-step process and, therefore, allowed a direct comparison between both processes.

Another compounding option for two-step processing was presented by means of a heating–cooling mixer (HCM). For the investigations, the type KM 23 HCM of the company Thyssen Henschel was used (Thyssen Henschel, now Zeppelin Systems, Kassel, Germany). A HCM is divided into two boilers with different temperature controls. The heating unit at 180 °C was used for compounding, whereas the cooling unit tempered at 20 °C was used for heat extraction from the melt and for pelletizing. This method makes it possible to use wood fibers instead of pellets as no continuous dosing is necessary. This process is referred to as “**2-step HCM**”.

The last compound option used was an internal mixer (IM). The mixing was performed with HAAKE Polylab OS Rheodrive combined with HAAKE Rheomix OS 3000 with Banbury rotors (ThermoFisher Scientific, Waltham, MA, USA). The processing temperature was set to 180 °C and the rotation speed was constantly kept at 20 rpm. After 10 min of mixing time, a steady torque was reached and the polymer was completely melted. The wood fibers were then added to the polymer melt. The internal mixer also makes it possible to use wood fibers as it is an intermittent process. The processing parameters chosen for the IM showed the lowest final temperature in the melt and the highest final fiber length in the study by Teuber [23]. The mixing was considered as complete when the torque and the temperature reached a steady level. This process is referred to as “**2-step IM**”.

Following all three two-step processes, the WPC parts were produced by means of an injection-molding machine from Wittmann Battenfeld (type HM160/750 S, WITTMANN Technology Ges.m.b.H., Vienna, Austria). To ensure comparability for all processes, the same barrel temperatures were set and the same injection mold with cold runner was used; in this case, it was a plate (290 mm × 190 mm × 2 mm). After production, the plates were stored at room temperature. Subsequently, tensile test specimens according to DIN EN ISO 527 type B—150 mm were milled out along the flow direction.

#### 2.2.2. Tests of Component Properties

Soxhlet extraction was used to extract the PP from the specimen using xylene as solvent in accordance with DIN EN ISO 10352. The remaining fibers were then dried and distributed on an optical scanner to analyze the fiber lengths of the wood fibers (type V850, Seiko Epson Corporation, Tokyo, Japan). The dispersed fibers were scanned and the scan was evaluated using the image analysis software FibreShape manufactured by IST AG, according to ISO 9276 (IST AG, Vilters, Switzerland).

Using differential scanning calorimetry (DSC), the wood-fiber plastic compounds were examined with regard to their behavior at elevated temperatures within an oxygen atmosphere. The oxidation induction time of a material quantifies its oxidation stability. The OIT was determined isothermally. The investigated materials were heated at 20 K/min in a nitrogen atmosphere. The temperature was kept constant at 180 °C for three minutes for equilibration. Next, the sample was exposed to an oxygen atmosphere. The method is explained in detail by Stammen [48]. The flow rate was 50 mL/min for both N_2_ and O_2_. The time interval between the first contact with oxygen and the start of oxidation defines the OIT, according to DIN EN ISO 11357. The tests were carried out using a DSC 2920 from TA Instruments (TA Instruments, New Castle, DE, USA).

The color values of the WPC surface were measured by a spectrophotometer (Byk-Gardner, Geretsried, Germany) (10° standard observer, D65 standard illuminate, color difference format ΔEab*), according to DIN EN ISO 11664. To assess the difference between two colors, a total color difference ΔE was determined (expressing the distance between two points in the L*a*b* system).

Headspace gas chromatography was performed to determine the specific amount of volatile organic compound (VOC) of the individual injection-molded parts in μg carbon per g sample, in accordance with VDA 277. The analyzed test specimens were sealed immediately after processing and until the measurement of the VOC was carried out. The samples were placed and sealed in headspace vials after calibration of the system. Next, they were tempered for 5 h at 120 °C in the headspace sampler so that all volatile components diffused into the gas phase and a stable equilibrium was established. A total of 0.05 mL of these volatiles was analyzed in the gas chromatograph (GC) using a flame ionization detector (FID). In order to do so, the volatiles were evaporated and transported through a column, along with the carrier gas (He 5.0). To evaporate the volatile components, the sample was isothermally exposed to a temperature of 50 °C for 3 min, then heated to 200 °C at 12 K/min and held at this temperature for 4.5 min. The FID recorded the individual spectrum, whose peaks were to determine the specific amount of VOC of the individual test specimens in μg carbon per g sample. A Perkin Elmer Inc. gas chromatograph of e Clarus 680 type was used (PerkinElmer, Inc., Waltham, MA, USA). Three molded parts of each processing method mentioned above were measured and packed airtight for the time between processing and measurement.

The tensile test, which was carried out according to ISO 527-2 to determine the tensile modulus and tensile strength of the milled specimens, was performed on a Zwick/Roell type Z020 universal testing machine (ZwickRoell GmbH & Co. KG, Ulm, Germany). The testing speed to determine the tensile modulus was set to 1 mm/min. To evaluate the tensile strength, a testing speed of 5 mm/min was chosen.

The microscopic images were taken with a Zeiss Smartzoom 5 digital microscope in coaxial bright field. Microsections were prepared with a polishing machine type Mecatech 250 SPC from Presi (Presi, Grenoble, France).

## 3. Results and Discussion

### 3.1. Fiber Length

Comparing the resulting fiber lengths of the injection-molded parts of the single-step and the two-step process (one-step IMC and two-step TSE), an 11% higher fiber length can be achieved by processing in just one step (Figure 3). After compounding via the IMC, the homogenized melt is merely pressed into the mold by means of a shotpot injection unit. The wood fibers were not reheated and mechanically stressed; this occurred in the two-step process. A comparison of all three two-step processes (two-step TSE, two-step HCM and two-step IM) shows that the HCM and IM result in lower fiber lengths. The mixing time within the heating cooling mixer and the internal mixer is significantly higher after more than 5 min. By compounding with the TSE, the wood fibers were fed into the melt after a length of 28 D, which resulted in a time under thermal and mechanical load of less than a minute. It is well known that the pelletizing process decreases the fiber length [29], but the results indicate that the time under thermal load has a greater impact. A comparison of the two-step HCM and two-step IM showed no significant difference in the arithmetic fiber length, although dry blending should result in a lower fiber length compared to feeding into the melt, as Teuber showed with wood flour [23]. The mixing time and time under load for both processes were similar. Furthermore, the analyses showed that the fiber length reduced by more than half across all the processes considered.

### 3.2. Oxidation Induction Time

In the case of wood fibers, the OIT can serve as a parameter for thermal damage. If compounding and injection molding are carried out gently, the measured OIT on the component drops, contrary to the usual interpretation. The components of the wood that lead to an earlier onset of the oxidation reaction, such as hemicellulose, do not degrade at this point and, therefore, appear in the measurement of the OIT. The higher the determined OIT of the wood composites, the more severe was the thermal damage that occurred in the previous processing steps. In general, the addition of wood drastically reduces the OIT of PP, which is also shown in the work of Borysiak [49].

The OIT of the investigated processing methods is shown in Figure 4. It is shown that an increased thermal degradation of the wood occurred in the single-step process. The OIT measured was 20 min. The reasons for this were the increased dwell time, shear stresses and pressures that occurred during processing. The intermediate melt reservoir, which acts as a buffer linking the continuous process of the twin-screw compounder with the discontinuous injection molding, increases the dwell time for the temperature-sensitive WPC. Furthermore, the wood-fiber plastic composite is exposed to high temperatures for longer periods of time compared to the two-step process. Additionally, direct processing requires a higher back pressure compared to the injection-molding machine, which also leads to an increased melting temperature. As a result, the wood fibers can thermally degrade further.

In comparison, the two-step processing using the twin-screw compounder and the injection-molding machine resulted in an OIT of 18 min. Compounding the WPC with the heating–cooling mixer resulted in an OIT of 17 min. Using the internal mixer to compound the WPC resulted in the lowest OIT (of 16 min) and, consequently, in the least chemical change in the wood. The geometry of the twin-screw compounder resulted in comparatively high thermal stress due to shear because of the small passage volume and gap dimensions of the cylinder. The shear force was considerably lower with suitable process control in the heating–cooling mixer. Furthermore, the wood fibers in the TSE were brought directly into contact with the hot plastic melt, whereas the materials of the composite were dry-blended in the HCM. The internal mixer offers the most gentle way of compounding the WPC when considering the result of the OIT of 16 min. The mixing chamber was only filled up to 60%; therefore the shear rate was reduced to a minimum. The thermal and mechanical damage caused by the shearing of the injection molding process was identical for both processes. The results of the OIT correlated with the discoloration of the tensile test bars, as shown in Figure 4 (right).

### 3.3. Volatile Organic Compounds

The occurrence of the discussed phenomena is supported by the investigations using headspace gas chromatography, which make it possible to determine the content of VOC of a material. The VOC includes hydrocarbons, alcohols and organic acids, which are released in the case of increased thermal damage [14,15,16,17,18,19]. The temperatures of around 180 °C occurring in the compounding process suggest that hemicellulose in particular degrades [12,13]. Accordingly, the measurement of emissions via headspace gas chromatography indicates the thermal damage sustained by materials. In Figure 5, the measured the emission values in μg carbon per g sample over the four process variants are shown. Since the test specimens were sealed immediately after processing and remained sealed until the VOC was measured, the measured values represent the emissions released due to the compounding rate. The single-step process shows twice the emission value (99.2 μgC/g) of the two-step processing with the twin-screw extruder and injection-molding machine (42.1 μgC/g.) Comparing all the two-step processing methods, the emission value can be reduced down to 6.3 μgC/g via compounding by means of an internal mixer. The measured emissions confirm the result of the OIT and the discoloration, discussed in Section 3.1.

To evaluate the influence of the different processing methods on the polymer itself, the neat PP resin was identically investigated. According to the measured results, it can be said that processing the polymer multiple times does increase the amount of measured μg carbon per gram sample. Therefore, the polymer itself experiences less stress when processed with the IMC. Furthermore, the results show that the different two-step melting processes themselves do not affect the VOC emission of neat PP in a significant way. This aligns with the comprehensive study by Xiang et al. [50], who showed that the amount of VOC on neat PP increases with multiple processing.

It is to be noted that the measured values for the WPC and for the PP resin could not be directly compared. In the applied measurement, the sample was heated, the volatile components evaporated and an equilibrium formed in the vapor space of the sample bottle. An aliquot of the vapor space above the sample was taken, injected directly into the FID and evaluated afterwards. However, the equilibrium of the volatile components depends strongly on the type and concentration of the analytes.

### 3.4. Tensile Properties

The tensile properties of the one-step processing by means of an IMC and the two-step processing with TSE were almost on the same level, as can be seen in Figure 6. The fiber length within in the one-step process was significantly higher, which should have led to increased mechanical properties [20,34,35], but the increased dwell time resulted in an elevated thermal load and seemed to deteriorate the benefits of the higher fiber length. A high dwell time was proven to be associated with reduced component mechanics in WPCs [21,22,25,26]. Increasing the throughput, thereby reducing the dwell time, could change the results [21]. Comparing the two-step processing methods, a clearly increased mechanical strength can be seen for the compounding method by means of the heating–cooling mixer. At 30 wt.-% of wood fiber content, a tensile strength of 42 MPa and tensile moduli of 3.5 GPa is possible. Compared to PP resin with a tensile modulus of 1.7 GPa and a tensile strength of 32 MPa, the stiffness can be doubled and the tensile strength can be improved by 30%. The compounding method by means of an internal mixer shows equal potential for high mechanical properties but lacks homogeneity at applied process control, as can be seen in the high standard deviation as well as in the cluster formations within the tensile bar depicted in Figure 7 (two-step IM). These investigations show that by increasing the mixing time, a homogeneous fiber distribution in the injection-molded part can be achieved and the mechanical properties e.g., the tensile strength, can be increased up to (43.3 ± 0.4) MPa. The fiber orientations, assessed qualitatively, are similar within all four of the tensile bars, as shown in Figure 7. Other reasons for the results of the tensile mechanics may include fiber–matrix adhesion and variations in the fiber content due to the gravimetric dosing caused by the non-uniform bulk density of the wood-fiber pellets. Based on the results, it can be said that, in the case of the wood fibers, the fiber length alone is not a decisive factor in the components’ mechanical performance.

The microsections of one tested tensile bar for each processing method were evaluated. The difference between the edge layer and the core layer can be clearly seen in each tensile bar. The fiber orientation was similar to that of the glass fibers, established by Menges [51]: in the flow direction in the edge layers and transverse to the flow direction in the low-shear core layer. It can be seen that the wood fibers did not have a uniform width, as in glass fibers. This can lead to the development of a weak point in the composite if the L/D ratios are insufficient [36,37].

## 4. Conclusions

In the field of lightweight construction, the application of wood fibers offers the potential to reduce the environmental impact of plastic composites by saving raw materials and energy. This requires ideal processing, starting with fiber production and compounding. This article compared four different compounding processes with regard to the resulting thermal load acting on the injection-molded wood–plastic composite. Not all processes are equally suitable for industrial applications. Continuous compounding methods, such as one-step IMC or two-step TSE, are preferable for applications with very high quantities. By choosing a gentle method of compounding, such as the use of an internal mixer, the least thermal damage occurs and, therefore, the best properties of the injection-molded part in terms of emissions and mechanical properties can be achieved, while homogeneity can be increased. Although the average fiber length resulting from the use of one-step processing is significantly higher when compared to two-step processes, the increased dwell time and, consequently, the elevated thermal load seem to damage the structural integrity of the wood fiber and counter the benefits of the enlarged fiber length. The findings of this article indicate that the structural integrity of wood fibers is more important in terms of their mechanical properties than the resulting fiber length.

## Figures and Tables

**Figure 1 materials-15-03393-f001:**
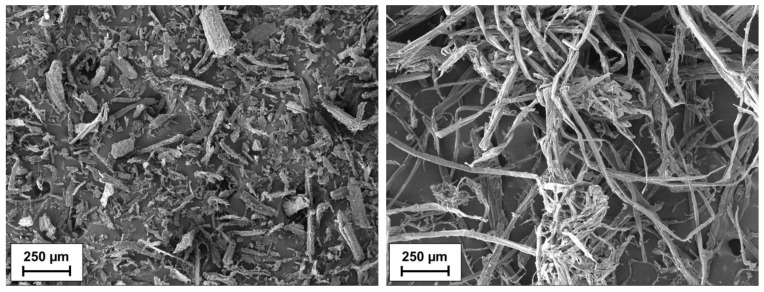
SEM backscatter images of wood flour (**left**) and wood fibers (**right**).

**Figure 2 materials-15-03393-f002:**
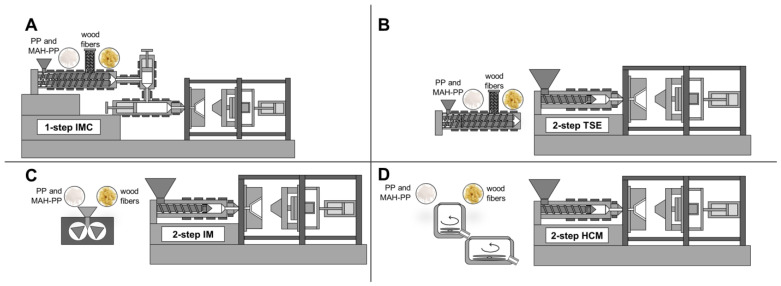
Schematic illustration of all melting processes investigated. (**A**) Injection-molding compounder, (**B**) twin-screw-extruder and injection molding, (**C**) internal mixer and injection molding, (**D**) heating cooling mixer and injection molding.

**Figure 3 materials-15-03393-f003:**
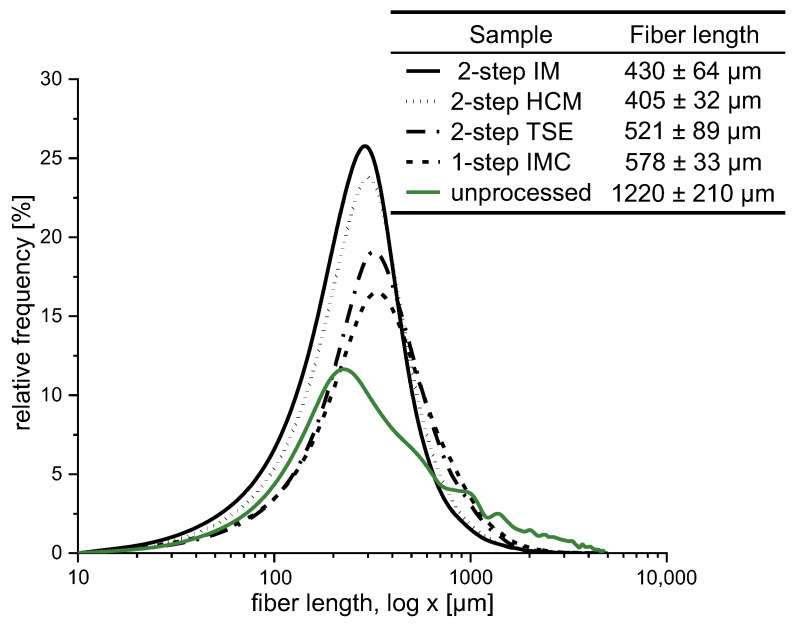
Residual arithmetic fiber length distribution (unrolled) in the component according to ISO 9276; *n* = 3; $\bar{X}\;\pm SD$; minimum number of measured objects >30,000.

**Figure 4 materials-15-03393-f004:**
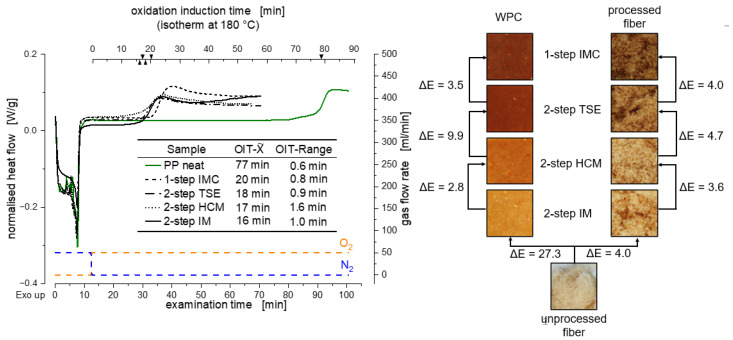
Comparison of the oxidation induction time of each individual melting process (measured on the WPC part) and the temperature effect during the processes shown on exemplary curve progressions (**left**); photographs of the tensile bars and extracted fibers show the discoloration due to processing quantified by the color deviation (**right**), *n* = 3; $\tilde{X}\;\mathrm{and}\;\mathrm{Range}$.

**Figure 5 materials-15-03393-f005:**
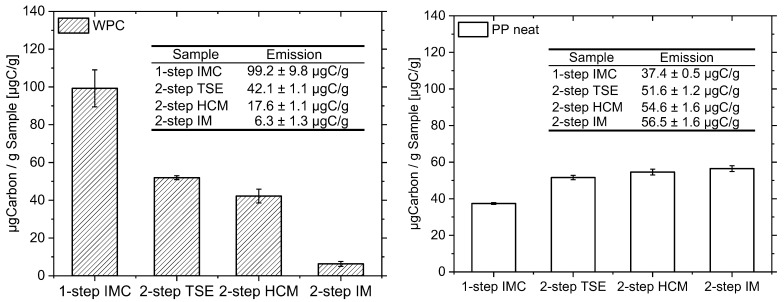
Resulting emissions in μgC/g for all processing methods analyzed; *n* = 9 for WPC (**left**), *n* = 3 for PP neat (**right**); $\bar{X}\;\pm SD$, normal distribution confirmed by Shapiro–Wilk test.

**Figure 6 materials-15-03393-f006:**
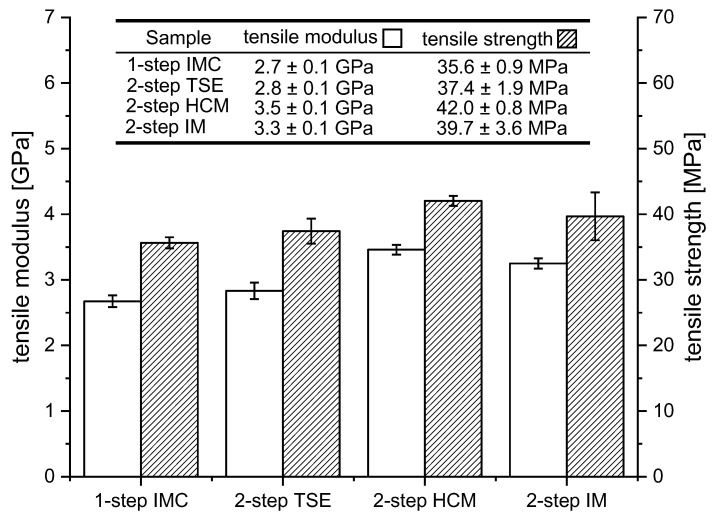
Tensile test according to ISO 527 at 30 wt.-% wood-fiber-reinforced PP; test specimens milled out of injection-molded sheets with s = 2 mm; *n* = 10; $\tilde{X}\;\pm SD$; values deviate between 10 and 20% from injection-molded test specimens, according to ISO 527. Normal distribution confirmed for every group of specimens by Shapiro–Wilk test.

**Figure 7 materials-15-03393-f007:**
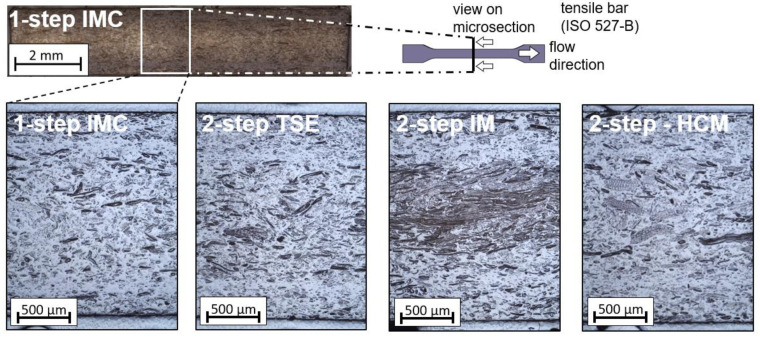
Microsection of the tested tensile rods in flow direction; position of the micrographs similar to those shown for one-step IMC; non-homogeneity and fiber bundles of the two-step IM process.

## Data Availability

The data presented in this study are available on request from the corresponding author.

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
