# Peer review of "Comparison of Melting Processes for WPC and the Resulting Differences in Thermal Damage, Emissions and Mechanics"

_materials, 2022, doi:10.3390/ma15093393_

Round 1
Reviewer 1 Report
The authors reported a detailed study about the comparison of melting processes for wood plastic composites. I recommend to accept this work after a minor revision.
1) Please provide the full name of WPC, wood plastic composites, for the first time.
2) In the Materials part, please provide the “Melt Flow Rate, MFR” of HJ120UB, this value is important for the melting process.
3) In the 2.2.2 part
In line 170: the testing speed should be 1 mm/min, as mentioned in ISO 527-2. Why did the authors choose the testing speed of 5 mm/min?
In line 181-207, according to VDA 277, “…at this temperature for 2.5 min …” (line 192) should be 4 min.
Besides, please provide the experimental information for SEM observation.
4) Figure 3: How to measure the fiber length?
“…..measured objects >30.000” (Line 228) means 30 or 30,000?
5) Figure 5: the melting process could affect the VOC emission of neat PP resins. Please provide the VOC results of neat PP resins after different melting processes in Figure 5. It is necessary to comparatively analyze the VOC emission of WPC prepared by different melting processes.
6) Figure 6:
It should be “.” rather than “,” for the decimal point.
7) Figure 7:
There is no A, B, C, and D on SEM images.
Reviewer 2 Report
The article "Comparison of melting processes for WPC - resulting differences in thermal damage, emissions and mechanics" discusses a significant degradation problem that polymer-wood composites are subjected to during technological processes in the melted state. The presented descriptions are correctly implemented; however, the work does not refer to the change of thermal properties and the analysis of the difference in the chemical composition of fillers and composites. If the article is to be accepted for publication, it must be supplemented. More detailed comments are provided below.
The article's title mentions thermal damage, but the authors did not specify the thermal stability of composites or fillers. In this case, the assessment of VOC and OIT emissions is insufficient and relates to the effects of degradation and not the stability of the materials themselves.
- Introduction evokes many references, although it does not allow for a clear definition of the main problems discussed in this paper. At the very beginning, the authors point to the thermal degradation of lignocellulosic fillers as the biggest problem, but they do not refer to this phenomenon later in the Introduction. The main degradation effects occurring during processing should be grouped and presented using the discussed methods, which will allow for a precise specification of the assumptions of the work performed.
- Information on the polymer used should be extended. It is a commercial material, but data such as MFI, density or applied modifiers should be specified. The reader did not have to search for a safety data sheet to perform a comparative analysis with other results.
It is strange that the authors did not wholly characterize the filler used after recalling various characteristics of wood-based fillers. The article should contain a separate section in work or in the supplementary data, in which the particle sizes, specific surface area, thermal stability, moisture content, chemical composition, and aspect ratio of the original natural fibers will be defined.
- The text should be corrected in terms of editing and adapting to the correct formatting of the journal. This applies, for example, to inconsistent use of indentation at the beginning of new paragraphs.
- The color analysis should include the study of the color of the filler itself, which will be subjected to thermal loads in time and temperature (under oxidizing conditions) comparable to the time the filler remains in the plasticizing system of the machines used, corresponding to the applied processing procedures. These data should be supplemented with FTIR spectroscopic assessment, and the changes in the chemical composition of the filler and composite induced by various thermomechanical and thermal loads should be defined, trying to correlate the processing methods with the predominant degradation effects.
- The research used did not present a reference sample made of PP only; it is quite a large deficiency that needs to be corrected.
- The order of the results discussed should be the same as the order of the methods described in the experimental part.
Round 2
Reviewer 2 Report
The authors supplemented the article and responded to the review, although not all comments were taken into account. However, considering the concepts of a limited fragment of the research topic in a compact form, the work can be accepted for publication.